# Harmony in Diversity: Improving All-in-One Image Restoration via Multi-Task Collaboration

Gang Wu
gwu@hit.edu.cn
Harbin Institute of Technology
Harbin, China

Junjun Jiang*
jiangjunjun@hit.edu.cn
Harbin Institute of Technology
Harbin, China

Kui Jiang
jiangkui@hit.edu.cn
Harbin Institute of Technology
Harbin, China

Xianming Liu
csxm@hit.edu.cn
Harbin Institute of Technology
Harbin, China

## ABSTRACT

Deep learning-based all-in-one image restoration methods have garnered significant attention in recent years due to capable of addressing multiple degradation tasks. These methods focus on extracting task-oriented information to guide the unified model and have achieved promising results through elaborate architecture design. They commonly adopt a simple mix training paradigm, and the proper optimization strategy for all-in-one tasks has been scarcely investigated. This oversight neglects the intricate relationships and potential conflicts among various restoration tasks, consequently leading to inconsistent optimization rhythms. In this paper, we extend and redefine the conventional all-in-one image restoration task as a multi-task learning problem and propose a straightforward yet effective active-reweighting strategy, dubbed **Art**, to harmonize the optimization of multiple degradation tasks. Art is a plug-and-play optimization strategy designed to mitigate hidden conflicts among multi-task optimization processes. Through extensive experiments on a diverse range of all-in-one image restoration settings, Art has been demonstrated to substantially enhance the performance of existing methods. When incorporated into the AirNet and TransWeather models, it achieves average improvements of **1.16** dB and **1.21** dB on PSNR, respectively. We hope this work will provide a principled framework for collaborating multiple tasks in all-in-one image restoration and pave the way for more efficient and effective restoration models, ultimately advancing the state-of-the-art in this critical research domain. Code and pre-trained models are available at our project page https://github.com/Aitical/Art.

## CCS CONCEPTS

• **Computing methodologies** → *Reconstruction*.

*Corresponding Author: Junjun Jiang

## KEYWORDS

Image Restoration, Multi-Task Learning, Image Dehazing, Image Deraining, Image Denoising

**ACM Reference Format:**
Gang Wu, Junjun Jiang, Kui Jiang, and Xianming Liu. 2024. Harmony in Diversity: Improving All-in-One Image Restoration via Multi-Task Collaboration. In *Proceedings of the 32nd ACM International Conference on Multimedia (MM '24), October 28-November 1, 2024, Melbourne, VIC, Australia.* ACM, New York, NY, USA, 9 pages. https://doi.org/10.1145/3664647.3680762

## 1 INTRODUCTION

In real-world scenarios, the observed images inevitably suffer from various types of degradation, such as noise, blur, rain, and haze [Karavarsamis et al. 2022]. These degradations deteriorate the visual quality of images and pose significant challenges to downstream tasks. Consequently, developing effective image restoration techniques [Ali et al. 2023] has become increasingly important for a wide range of applications, and been a fundamental research topic in recent decades [Banham and Katsaggelos 1997; Su et al. 2022].

Recently, deep-learning based single-task models have demonstrated exceptional performance in the respective domains, such as denoising [Liang et al. 2021; Zamir et al. 2022; Zhang et al. 2023b], deblurring [Chen et al. 2022a; Pham et al. 2024; Wang et al. 2022], deraining [Chen et al. 2023, 2024; Jiang et al. 2020; Zamir et al. 2021], and dehazing [Cui et al. 2023; Qin et al. 2020]. However, these task-specific methods face a significant challenge in real-world scenarios: when they excel in a certain task, they may completely fail in other tasks. In this paper, we focus on the all-in-one image restoration, which can integrate knowledge across a collection of related tasks, and propose to refresh the performance of existing all-in-one methods by the propose Art, as shown in Figure 1.

Beyond the single-task model illustrated in Figure 2 (a), there has been growing research interest in developing unified models to handle multiple degradation tasks within a single model, thereby providing more comprehensive and practical solutions to real-world image restoration challenges [Chen et al. 2021; Li et al. 2022; Liu et al. 2022; Potlapalli et al. 2023; Valanarasu et al. 2022; Zhang et al. 2023a]. In [Chen et al. 2021], an early attempt to address multiple tasks within a single training process introduces a multi-head and multi-output architecture with a shared backbone, as illustrated in Figure 2 (b). Recent research has shifted attention to unified models, which is achieved by harnessing task-oriented information

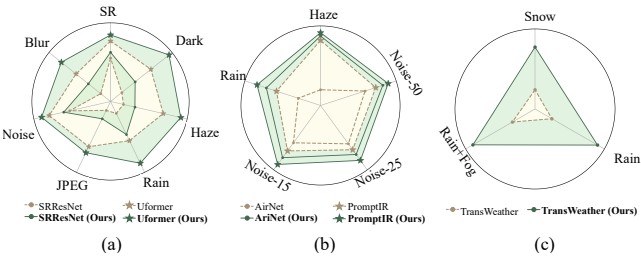

**Figure 1: Comparative analysis of the proposed Art approach against SOTA all-in-one image restoration methods. When integrating Art to retrain existing models, it consistently enhances the performance of the previous ones across diverse all-in-one image restoration tasks, underscoring its efficacy and versatility in handling multiple degradation tasks.**

to inform and guide the learning process of the unified model [Li et al. 2022; Liu et al. 2022; Zhang et al. 2023a]. In [Li et al. 2022], the authors introduce a self-supervised approach [He et al. 2020] to pretrain a degradation encoder, which is fixed and utilized to extract the degradation representations of the input for the training of the all-in-one model. The recent PromptIR [Potlapalli et al. 2023] and Prompt-In-Prompt [Li et al. 2023] further simplify the task-oriented guidance by an adaptive prompt learning approach, enabling end-to-end multi-task training and achieving robust performance. With the development of ingenious architectures, significant strides have been made in these methods. Most recently, MioIR [Kong et al. 2024] adopts a novel training scheme to improve the all-in-one image restoration problem by a sequential and prompt learning strategy.

However, we observe that these methods typically adopt mixed training, which involves combining the optimization of individual tasks together. Less attention is paid to the optimization process of all-in-one tasks, thereby overlooking the intricate relationships and potential conflicts among multiple degradations in the mixed training paradigm. Although MioIR [Kong et al. 2024] has improved the performance of all-in-one image restoration by elaborately changing the task settings, it fundamentally does not solve the conflict between tasks. To better illustrate this insight, we present the loss curves of a 7-task all-in-one training process in Figure 2 (c). It is evident that large oscillations exist during training, and inconsistent convergence occurs for different tasks. We argue that this uncertain dynamics in the training process is due to hidden conflicts and inconsistent optimization objectives between multiple degraded tasks. Therefore, failing to account for these drawbacks through simply mix training can lead to performance discrepancies of different degradation tasks and hinder the overall convergence for the all-in-one restoration model. This underscores the need for a more sophisticated approach to handle the intricate relationships among different restoration tasks and improve all-in-one image restoration models through multi-task collaboration.

To address this challenge, we propose an enhanced all-in-one image restoration approach via multi-task reweighting, denoted as **Art**. As illustrated in Figure 1 (d), we extend the loss function of all-in-one tasks with active-reweighting for each individual task, and one can observe that retrained all-in-one model with the proposed Art achieves a more stable learning process. Specifically, Art

explicitly formulates the restoration process as a multi-task learning problem, aiming to harmonize the optimization of multiple restoration tasks. To achieve this goal, we explore the convergence status of each individual task from its local and global optimization process and propose an active-reweighting strategy, which can equilibrate the optimization among different restoration tasks. In detail, this reweighting approach comprises two key components: a local convergence rate (LCR) and a global correction weight (GCW). The former addresses the challenge of inconsistent convergence within a single iteration, while the latter dynamically adjusts the optimization objective when a task deviates from its best performance. The proposed Art is straightforward yet effective in mitigating the detrimental effects of hidden conflicts and ensures a delicate balance among individual tasks. Notably, Art is a generalized optimization strategy that can be applied as a plug-and-play method to existing all-in-one approaches.

The current landscape of all-in-one image restoration research is characterized by a diverse range of task settings and model architectures. To validate the efficacy of our approach, we have summarized several prevalent all-in-one experimental settings. For each experimental setting, we retrained the previous models with our Art approach. When our approach is incorporated into the AirNet and TransWeather methods, it achieves average improvements of **1.16** dB and **1.24** dB on PSNR, respectively. As depicted in Figure 1, the results consistently demonstrate that our proposed Art yields significant improvements in various task settings and model architectures, highlighting its robustness and generalizability.

The main contributions of our work can be summarized as follows:

- We revisit all-in-one image restoration from the perspective of multi-task learning and propose a novel active approach, Art, that explicitly addresses the challenges of inconsistency and potential conflicts among multiple restoration tasks.
- We propose a straightforward yet effective loss function with task-specific reweighting, comprising a local convergence rate and a global correction weight. These components work together to prevent the inhibition of multi-task learning and dynamically adjust the optimization objective, ensuring a delicate balance between individual task progress and overall model performance.
- We demonstrate the effectiveness of Art by validating its efficacy across diverse all-in-one tasks with varying numbers of degradation types. The consistent improvements observed in different task settings and architectures highlight the potential of Art to become a promising optimization strategy in all-in-one image restoration.

## 2 THE PROPOSED METHOD

### 2.1 All-in-one Image Restoration

All-in-one image restoration aims to recover clean images from corrupted observations that suffer from multiple degradations simultaneously. Given a set of degraded images x and their corresponding high-quality ground truths y, the goal is to learn a restoration model $f_\theta$ parameterized by $\theta$ that can effectively map the degraded inputs to the desired clean outputs. Mathematically, the optimization process of all-in-one image restoration can be formulated as:

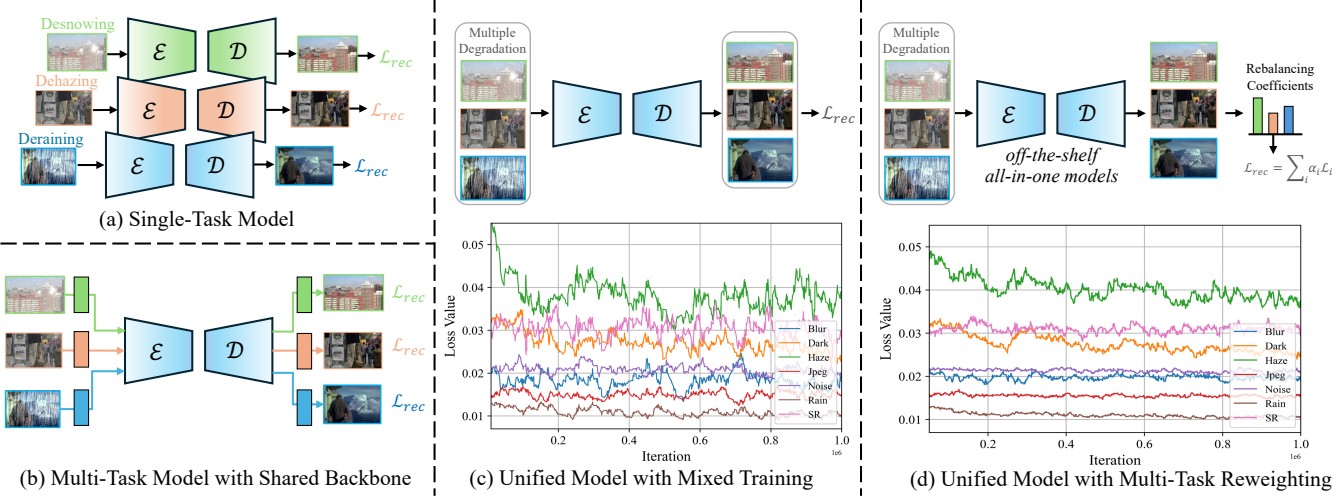

**Figure 2: Evolution of image restoration methods and convergence comparison between mixed training and our approach. (a) End-to-end training of single-task image restoration models. (b) Multi-task image restoration model with a shared backbone [Chen et al. 2021]. The model is trained separately for each task using a shared backbone and task-specific input and output heads, limiting its applicability. (c) Top: Architecture of current all-in-one methods, aiming to obtain a unified model through mixed training [Li et al. 2022; Potlapalli et al. 2023; Valanarasu et al. 2022]. Bottom: Loss curves during the training process, where collisions are observed in multi-task optimization. (d) Top: Our proposed Art, explicitly addressing all-in-one image restoration as a multi-task problem. Bottom: Loss curves of our approach, achieving a more robust learning process.**

$$\theta^* = \min_\theta L(f_\theta(x), y), \tag{1}$$

where $L$ is a reconstruction loss, such as the Mean Absolute Error (MAE), which measures the difference between the restored images $f_\theta(x)$ and the ground truths y. The model parameters $\theta$ are optimized through an end-to-end training process to minimize the reconstruction loss.

## 2.2 Multi-Task Collaboration

To address the challenges of hidden conflict in multiple degradation types, we extend the loss function of all-in-one image restoration tasks with explicit active-reweighting. In detail, we assume that the degradation type of each training sample is known, and we denote the input as $x_i^t$, where $i \in 1, \ldots, N$ represents the sample index, $t \in 1, \ldots, K$ indicates the degradation type, $N$ is the total number of samples, and $K$ is the number of degradation types.

We introduce a simple but effective modification of conventional loss function, which brings significant improvements to the optimization process. Specifically, we reformulate the objective function as:

$$L = \sum_t \alpha^t L^t, \tag{2}$$

where $\alpha^t$ is an adaptive rebalancing coefficient for each degradation type, and $L^t = L(\mathcal{X}^t, \mathcal{Y}^t)$ represents the loss of individual task $t$. $\mathcal{X}^t$ and $\mathcal{Y}^t$ are the subsets of degraded images and their corresponding ground truths with degradation type $t$, respectively.

When the parameter weights $\theta$ are updated by the chain rule of backpropagation, we obtain:

$$\frac{\partial L}{\partial \theta} = \sum_t \alpha^t \frac{\partial L^t}{\partial \theta}, \tag{3}$$

The adaptive reweighting coefficient $\alpha^t$ offers a straightforward approach to control the optimization of each task within the explicitly formulated multi-task loss function. As observed in Section 1, when a biased task is identified, increasing its corresponding task weight $\alpha^t$ can compel the parameter weights $\theta$ to rectify the solution. The primary challenge lies in determining the appropriate task weight $\alpha^t$ to adaptively balance the tasks. To address this, we propose a novel approach that examines the loss value of each individual task from two perspectives: its current convergence status and its global optimization direction.

The local convergence rate provides valuable insights into the optimization progress of each task within a single iteration. By analyzing the rate of change in the loss value, we can identify tasks that are converging slower than others and adjust their corresponding task weights accordingly. This local perspective ensures that all tasks progress at a more consistent pace, preventing any single task from dominating the optimization process.

Furthermore, we introduce a global optimization direction component that considers the overall trajectory of each task's loss value across historical iterations. This global perspective allows us to detect tasks that deviate significantly from their optimal path and dynamically adjust their task weights to steer them back towards their desired objectives. By incorporating this global correction mechanism, we can effectively mitigate the detrimental effects of conflicting gradients and promote a more harmonious optimization process.

The final task weight $\alpha^t$ is obtained by combining the insights gained from both the current convergence status and the global optimization direction. This adaptive reweighting scheme enables our proposed method to dynamically adjust the influence of each task on the overall optimization process, ensuring a delicate balance between individual task progress and the collective performance of the multi-task learning model.

In the following subsections, we delve into the details of computing the current convergence rate and the global optimization direction, and how they are seamlessly integrated to derive the adaptive rebalancing coefficient $\alpha^t$. Through this meticulous approach, our method effectively addresses the challenges of multi-task learning in all-in-one image restoration, paving the way for more efficient and balanced optimization processes.

*local convergence rate*. The local convergence rate factor $s^t$ is introduced to balance the convergence rates of different restoration tasks. It is computed as the ratio between the current iteration loss $L_t^{iter}$ and the previous iteration loss $L_t^{iter-1}$ for each degradation type $t$:

$$s^t = L_t^{iter}/L_t^{iter-1}. \qquad (4)$$

The convergence rate factor $s^t$ is instrumental in addressing biased tasks by dynamically adjusting the optimization speed of each task based on its convergence behavior. When a task's loss begins to increase, signaling a deviation from the desired convergence trajectory, $s^t$ assumes a value greater than 1. This elevated value effectively amplifies the gradient weight of the corresponding task, compelling the optimization process to prioritize the correction of the suboptimal convergence. Moreover, there is potential influence of sample quality in mini-batch training, which can give rise to unreliable coefficients, and we propose a more robust convergence rate estimate with an exponential moving average (EMA) strategy. This approach involves smoothing the historical loss values to attenuate the impact of local fluctuations and anomalies. The smoothed convergence rate, denoted as $\tilde{s}^t$, is formulated as follows:

$$\tilde{s}^t = L_t^{iter}/\tilde{L}_t, \qquad (5)$$

where $L_t^{iter}$ and $\tilde{L}_t$ are loss values for the current and the smoothed one, respectively. The smoothed loss values are obtained using the EMA:

$$\tilde{L}_t = \beta\tilde{L}_t + (1-\beta)L_t^{iter}, \qquad (6)$$

where $\beta \in [0,1]$ is a hyperparameter that controls the degree of smoothing. A larger $\beta$ value gives more weight to the historical losses, resulting in a smoother convergence rate factor.

The smoothed loss value reflects the overall convergence state of the target task, allowing the local convergence rate to steadily capture the current convergence status.

*global correction weight*. The global correction factor $r^t$ is introduced to mitigate inter-task conflicts and prevent the performance degradation of individual tasks. It is computed as the ratio between the historical minimum loss $L_t^{min}$ and the current iteration loss $L_t^{iter}$ for each degradation type $t$:

$$r^t = L_t^{iter}/L_t^{min}, \qquad (7)$$

**Table 1: Effectiveness of the proposed local convergence rate and global correction weight.**

| $\tilde{s}$ | | $\tilde{r}$ | | PSNR |
|---|---|---|---|---|
| $w/o$ | $w$ | $w/o$ | $w$ | |
| | | | | 29.52 |
| ✓ | | | | 29.66 |
| | ✓ | | | 29.81 |
| | | ✓ | | 14.77 |
| | | | ✓ | 29.71 |
| | ✓ | | ✓ | 30.05 |

The effectiveness of the global correction factor lies in its ability to dynamically adjust the task weights based on their relative performance. When a task's current loss is much higher than its historical minimum, $r^t$ becomes greater than 1, indicating the necessity for increased attention to that task. By allocating more weight to the biased task, the global correction factor facilitates the steering of the optimization process towards a more robust solution.

Moreover, it is crucial to recognize that directly utilizing the global correction factor can potentially introduce instability issues, akin to the local oscillations observed in the loss values. To address this concern, we propose the application of a logarithmic transformation to the current and minimal loss values, yielding a more stable result denoted as $\tilde{r}^t$.

*Active Reweighting*. To effectively manage the optimization of multiple degradation types within our multi-task framework, we have developed a novel strategy for adaptive task weighting $\alpha^t$. This approach strategically integrates two key components: the local convergence rate, $\tilde{s}^t$, and the global correction weight, $\tilde{r}^t$. The combination of these factors enables a balanced and stable optimization process across various tasks.

The adaptive task weight $\alpha^t$ is mathematically formulated as follows:

$$\alpha^t = \frac{\exp(\tilde{s}^t \cdot \tilde{r}^t/\tau)}{\sum_{j=1}^K \exp(\tilde{s}^j \cdot \tilde{r}^j/\tau)}, \qquad (8)$$

where $\tau$ represents the temperature coefficient. By leveraging these adaptive weights, our framework can more effectively prioritize tasks that require attention, thereby enhancing the overall efficacy of the multi-task learning process.

### 2.3 Remark

Our proposed active-reweighting strategy addresses a critical challenge in all-in-one image restoration: the hidden conflicts in multi-task learning. The core of our approach is the adaptive coefficient $\alpha$, composed of two key elements: the local convergence rate (LCR), $\tilde{s}$, and the global correction weight (GCW), $\tilde{r}$. These components work in tandem to dynamically balance the optimization process across multiple tasks.

Table 1 presents our ablation study, demonstrating the effectiveness of each component. The baseline mixed training achieves a PSNR of 29.52 dB. Introducing LCR without smoothing improves PSNR to 29.66 dB, while smoothed LCR further enhances performance to 29.81 dB. This improvement underscores the importance of stabilizing convergence rates across tasks. The GCW proves

**Table 2: Comprehensive evaluation of the proposed Art framework across diverse experimental settings in existing all-in-one image restoration research. As a general optimization strategy, Art is assessed on various all-in-one image restoration tasks and models.**

| Experiment Settings | Count of Tasks | Detail Degradation |
|---|---|---|
| *multiple degradation* [Kong et al. 2024] | 7 | SR, Blur, Noise, JPEG, Rain, Haze, Low-Light |
| *rain-haze-noise* [Li et al. 2022] | 5 | Rain, Haze, Noise-$\sigma$15, Noise-$\sigma$25, Noise-$\sigma$50 |
| *rain-haze-snow* [Valanarasu et al. 2022] | 3 | Rain, Haze, Snow |
| *rain-haze-noise-blur-dark* [Zhang et al. 2023a] | 5 | Rain, Haze, Noise, Blur, Low-Light |

**Table 3: Ablation study on weight of EMA $\beta$**

| $\beta$. | 0 | 0.001 | 0.01 | 0.1 |
|---|---|---|---|---|
| PSNR | 29.67 | **29.81** | **29.81** | 29.79 |

**Table 4: Ablation study on weight of EMA $\beta$**

**Table 5: Ablation study on temperature $\tau$.**

| $\tau$ | 0.1 | 1 | 3 | 7 | 10 |
|---|---|---|---|---|---|
| PSNR | 29.59 | 29.87 | **30.05** | 29.99 | 29.93 |

crucial as well. Collapse of the GCW leads to severe degradation (14.77 dB PSNR), while the log-transformed GCW significantly improves PSNR to 29.71 dB. The synergistic combination of smoothed LCR and log-transformed GCW achieves optimal performance at 30.05 dB PSNR, surpassing all other configurations. These results highlight our method's ability to stabilize convergence rates across tasks, prevent learning collapse, address task bias, and dynamically adjust task weights based on both local and global optimization behaviors.

## 3 EXPERIMENT

### 3.1 Experiment Setting

In this paper, we conduct extensive experiments to evaluate the effectiveness and generalization of our proposed Art framework. Considering core contribution of our Art is the rebalanced loss function, which is a plug-and-play component to existing models, we take four different experimental settings, covering a wide range of image restoration tasks and datasets, as presented in Table 2. Detailed experimental settings are as follows:

i. **'multiple degradation' setting**: Following [Kong et al. 2024], there are 7 different image restoration tasks. The multi-degradation dataset is synthesized based on DF2K dataset, which provides the unbiased analysis of influence between tasks.

ii. **'rain-haze-noise' setting**: Following [Li et al. 2022; Potlapalli et al. 2023], the experiments span a range of datasets and contain 5 different degradation types. Specifically, BSD400 and WED datasets are combined for training image denoising, with testing on BSD68. Noisy images are artificially created using white Gaussian noise at 15, 25, and 50 levels. Rain100L is utilized for deraining, and RESIDE, which comprises the outdoor training set (OTS) and the synthetic outdoor test dataset (SOTS), is used for image dehazing.

iii. **'rain-haze-snow' setting**: Following [Valanarasu et al. 2022], the training dataset, termed "AllWeather", includes images

from Snow100K [Liu et al. 2018], Raindrop [Qian et al. 2018], and Outdoor-Rain [Li et al. 2019a], encompassing a variety of weather-related degradations. The testing is carried out on synthetic and real-world datasets, including Test1 [Li et al. 2019a], RainDrop [Qian et al. 2018], and Snow100k-L test datasets.

iv. **'rain-haze-noise-blur-dark' setting**: Following [Zhang et al. 2023a], there are 5 different tasks. Specifically, we employ Rain100L [Yang et al. 2017] for deraining, the Indoor Training Set (ITS) from the RESIDE [Li et al. 2019b] dataset for dehazing, a combination of BSD400 [Arbeláez et al. 2011] and WED [Ma et al. 2017] for denoising, GoPro [Nah et al. 2017] for deblurring, and LOL [Wei et al. 2018] for low-light enhancement.

For all experimental settings described above, we take the corresponding state-of-the-art methods as our baselines. We strictly follow their original training configurations to ensure a fair comparison. Additionally, we retrain these baseline models using their official code to adapt to our experimental settings and provide a through evaluation of the effectiveness of our proposed Art.

### 3.2 Ablation Studies

Center of the our Art framework is the proposed rebalanced loss function, which is straightforward yet effective. There are only an adaptive rebalancing weight is involved beyond existing methods. Here we want to discuss some other components and provide more detailed insights of our Art. Detailed ablation studies are provides as follows.

**Impact of EMA weight.** To evaluate the impact of the weight $\beta$ in our proposed method, we conduct an ablation study with different values of $\beta$ ranging from 0 to 0.1. The results are presented in Table 4. Notably, to better understand the influence of $\beta$, this study is conducted with only $\tilde{s}$ instead of $\alpha$, because the $\beta$ only works with $\tilde{s}$. When $\beta$ is set to 0, representing no EMA component, the PSNR reaches 29.67 dB. This serves as a baseline for evaluating the contribution of the EMA in our method. Setting $\beta$ to 0.01 and 0.001 yields a PSNR of 29.81 dB in both cases. This suggests that EMA strategy works effectively to capture the historical information and stabilize the training process, highlighting the contribution of our proposed local convergence rate. Finally, as we increase $\beta$ to 0.1, the PSNR induces to 29.79 dB.

**Impact of Temperature coefficient.** To investigate the influence of the temperature parameter $\tau$ on our proposed method's performance, we conduct an ablation study by varying the values of $\tau$ from 0.1 to 10. The results are presented in Table 5. When $\tau$

**Table 6: Comparison on 7 distinct degradation tasks introduced in [Kong et al. 2024].**

| Methods | SR | Blur | Noise | JPEG | Rain | Haze | Low-Light | Avg. |
|---|---|---|---|---|---|---|---|---|
| SRResNet-M | 25.52 | 30.01 | 30.49 | 32.46 | 32.38 | 25.57 | 30.20 | 29.52 |
| SRResNet-S [Kong et al. 2024] | 25.72 | 30.49 | 30.67 | 32.73 | 32.81 | 25.78 | 30.45 | 29.84 |
| **SRResNet-M + Art (Ours)** | **25.59** | **30.31** | **30.59** | **32.61** | **33.24** | **26.30** | **31.71** | **30.05** |
| **SRResNet-S + Art (Ours)** | **25.78** | **30.78** | **30.76** | **32.85** | **33.69** | **26.51** | **32.04** | **30.34** |
| Uformer-M | 25.80 | 30.53 | 30.84 | 33.13 | 33.39 | 27.93 | 33.27 | 30.70 |
| Uformer-S [Kong et al. 2024] | 26.07 | 31.11 | 30.96 | 33.27 | 35.96 | 28.29 | 32.80 | 31.21 |
| **Uformer-M + Art (Ours)** | **25.90** | **30.84** | **30.97** | **33.24** | **34.26** | **28.97** | **35.18** | **31.34** |
| **Uformer-S + Art (Ours)** | **26.16** | **31.50** | **31.16** | **33.40** | **36.54** | **29.48** | **34.40** | **31.81** |

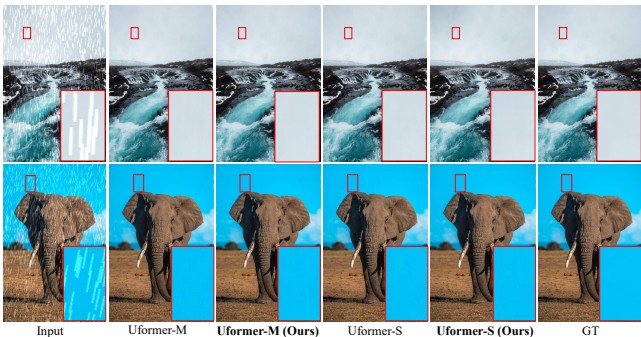

Input    Uformer-M    **Uformer-M (Ours)**    Uformer-S    **Uformer-S (Ours)**    GT

**Figure 3: Visual results of image deraining task. Compared to baseline all-in-one Uformer model introduced in [Kong et al. 2024], our proposed Art can consistently enhance the existing models even with the sequential training approach.**

is set to a small value of 0.1, the PSNR reaches 29.59 dB. However, we observe that this value is too small to obtain a robust $\alpha$, leading to oscillations during the training process. As we increase $\tau$ to 1, the PSNR improves to 29.87 dB, indicating that a higher temperature facilitates better optimization and convergence. The best performance is achieved when $\tau$ is set to 3, with a PSNR of 30.05 dB. This suggests that an appropriate balance is struck among the multiple tasks at this temperature value, enabling the model to effectively learn and adapt to the different task requirements. Further increasing $\tau$ beyond 3 leads to a slight decline in performance. This behavior can be attributed to the fact that higher values of $\tau$ provide a more uniform weight distribution among the multiple tasks, potentially impeding the flexibility of our adaptive rebalancing weight. Consequently, the model's ability to dynamically adjust the task weights based on their individual performance is hindered. It is worth noting that even with suboptimal hyper-parameter settings, our proposed method still outperforms the baseline models presented in [Kong et al. 2024], which achieve a PSNR of 29.51 dB. This demonstrates the robustness and effectiveness of our approach in handling multi-task learning scenarios, even when the temperature parameter is not optimally tuned.

## 3.3 Main Results

*'multiple degradation' setting*. Table 6 demonstrates the superior performance of our proposed multi-task balanced learning approach, Art, compared to the vanilla mixed Training and the state-of-the-art sequential training strategy introduced by MiOIR

[Kong et al. 2024]. The sequential training strategy, denoted as -S in the table, aims to mitigate the interference between different tasks by gradually increasing the number of tasks in the training set. While this approach has been shown to improve upon the mixed training paradigm in all-in-one image restoration, it has limitations in terms of scalability to multiple tasks due to its sensitivity to the training order. In contrast, our Art method achieves significant improvements over the sequential training strategy using the most straightforward mixed training approach. By incorporating task-specific rebalancing coefficients, Art effectively equilibrates dynamics optimization and mitigates inter-task conflicts and surpasses the performance of the carefully designed sequential training strategy without the need for controlling the task order.

Furthermore, we observe that the sequential training strategy introduced in MiOIR ultimately reduces to mixed training when all tasks are included in the final stage. Building upon this insight, we apply our multi-task reweighting strategy to the mixed training portion of the sequential training pipeline. The results demonstrate that our approach can further enhance the performance of sequential training, achieving notable improvements.

Figure 3 presents a subjective comparison of the restored images, showcasing the superior image quality achieved by our retrained SRResNet model. The visual improvements are evident not only when compared to the baseline models but also when contrasted with the sequential training approach. This further highlights the effectiveness of our multi-task balanced learning method in enhancing the perceptual quality of the restored images.

*"dhaze-derain-denoise" setting*. Table 7 presents the experimental results for three distinct degradation tasks: rain, haze, and noise, which are widely investigated in the current research landscape of all-in-one image restoration. By employing our proposed method, Art, to retrain the AirNet and PromptIR models, we observe significant improvements across all tasks, highlighting the effectiveness of our approach in addressing the challenges of multi-task imbalance.

Notably, the retrained AirNet model using Art achieves significant improvement of performance that are comparable to, or even surpass, results of the original PromptIR model. This finding directly corroborates the severe impact of multi-task imbalance on the current state of all-in-one image restoration. The original AirNet model struggles to attain optimal results due to the inherent difficulties in balancing multiple tasks during training. However, by applying our Art approach, we effectively mitigate these issues and

**Table 7: Comparative results with rain-haze-noise all-in-one restoration tasks. One can find that our retrained AirNet and PromptIR outperforms original ones on all test datasets.**

| Methods | Dehazing | Deraining | Denoising on BSD68 dataset | | | Average |
| --- | --- | --- | --- | --- | --- | --- |
| | SOTS | Rain100L | $\sigma = 15$ | $\sigma = 25$ | $\sigma = 50$ | |
| FDGAN [Dong et al. 2020] | 24.71/0.924 | 29.89/0.933 | 30.25/0.910 | 28.81/0.868 | 26.43/0.776 | 28.02/0.883 |
| MPRNet [Zamir et al. 2021] | 25.28/0.954 | 33.57/0.954 | 33.54/0.927 | 30.89/0.880 | 27.56/0.779 | 30.17/0.899 |
| AirNet [Li et al. 2022] | 27.94/0.962 | 34.90/0.967 | 33.92/0.933 | 31.26/0.888 | 28.00/0.797 | 31.20/0.910 |
| **AirNet + Art (Ours)** | **30.56/0.977** | **37.74/0.981** | **34.02/0.934** | **31.37/0.890** | **28.12/0.802** | **32.36/0.917** |
| PromptIR [Potlapalli et al. 2023] | 30.58/0.974 | 36.37/0.972 | 33.98/0.933 | 31.31/0.888 | 28.06/0.799 | 32.06/0.913 |
| **PromptIR + Art (Ours)** | **30.83/0.979** | **37.94/0.982** | **34.06/0.934** | **31.42/0.891** | **28.14/0.801** | **32.49/0.917** |

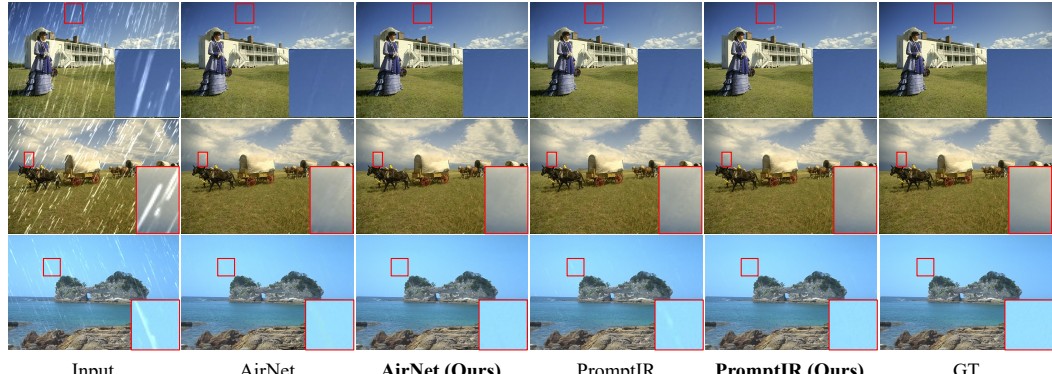

|  Input  |  AirNet  |  **AirNet (Ours)**  |  PromptIR  |  **PromptIR (Ours)**  |  GT  |

**Figure 4: Visual results of AirNet, PromptIR and retrained ones by the propsoed Art.**

**Table 8: Performance evaluation of deweathering all-in-one tasks. Consistent improvement is achieved with the proposed approach.**

| Datasets | Method | PSNR ↑ | SSIM ↑ |
| --- | --- | --- | --- |
| | MPRNet [Zamir et al. 2021] | 28.03 | 0.9192 |
| | All-in-One [Li et al. 2020] | 24.71 | 0.8980 |
| Outdoor-Rain | WeatherDiff$_{128}$[Özdenizci and Legenstein 2023] | 29.72 | 0.9216 |
| | TransWeather [Valanarasu et al. 2022] | 28.83 | 0.9000 |
| | **TransWeather + Art (Ours)** | **29.81** | **0.9088** |
| | DDMSNet [Zhang et al. 2021] | 28.85 | 0.8772 |
| | All-in-One [Li et al. 2020] | 28.33 | 0.8820 |
| Snow100K | WeatherDiff$_{128}$[Özdenizci and Legenstein 2023] | 29.58 | 0.8941 |
| | TransWeather [Valanarasu et al. 2022] | 29.31 | 0.8879 |
| | **TransWeather + Art (Ours)** | **30.61** | **0.9083** |
| | IDT [Xiao et al. 2023] | 31.87 | 0.9313 |
| | All-in-One [Li et al. 2020] | 31.12 | 0.9268 |
| RainDrop | WeatherDiff$_{128}$[Özdenizci and Legenstein 2023] | 29.66 | 0.9225 |
| | TransWeather [Valanarasu et al. 2022] | 30.17 | 0.9157 |
| | **TransWeather + Art (Ours)** | **31.54** | **0.9338** |
| Average | TransWeather [Valanarasu et al. 2022] | 29.44 | 0.9012 |
| | **TransWeather + Art (Ours)** | **30.65** | **0.9170** |

unlock the true potential of the AirNet architecture, demonstrating its capacity to achieve state-of-the-art performance when properly optimized.

Conversely, the PromptIR model, with its large model capacity (near 36M parameters) and adaptive prompt mechanism, exhibits strong generalization capabilities across multiple tasks. The adaptive prompt enables the model to dynamically adjust its behavior based on the specific characteristics of each task, leading to improved performance. However, despite its advanced architecture, the PromptIR model is not impervious to the inherent conflicts and imbalances that arise in multi-task learning when employing vanilla

mixed training approaches. By retraining the PromptIR with our Art method, we observe substantial enhancements in restoration quality across all three degradation tasks as well. The application of the proposed task-specific rebalancing coefficients effectively equilibrates the dynamics optimization and mitigates inter-task conflicts. This enables the retrained PromptIR model to surpass its original counterpart and provides a new benchmark in all-in-one image restoration.

***"dhaze-derain-desnow" setting***. Table 8 showcases the performance of our Art method applied to the TransWeather. Compared to the original model, the retrained version exhibits average PSNR gains of 1.21 dB. These improvements highlight the enhanced ability of the retrained model to restore weather-degraded images. Figure 5 provides visual comparisons, corroborating the quantitative results.

Moreover, Table 9 presents results on real-world deweathering tasks. Notably, the proposed Art method significantly improves the TransWeather model's performance, elevating it to a level comparable with state-of-the-art methods in terms of average performance across various real-world scenarios.

The alignment between quantitative PSNR improvements and qualitative visual enhancements offers compelling evidence of our approach's efficacy in addressing various weather-related image degradations.

***"rain-haze-snow-blur-dark" setting***. Table 10 presents a comprehensive comparison of our proposed Art method applied to the AirNet and Transweather models for all-in-one image restoration across five distinct tasks. We evaluate the performance of the retrained models against their original counterparts and the results

**Table 9: Comparison on real-world deweathering datasets proposed in [Zhu et al. 2023].**

| Methods | Rain on SPA+ | | Snow on RealSnow | | Haze on REVIDE | | Average | |
|---|---|---|---|---|---|---|---|---|
| | PSNR | SSIM | PSNR | SSIM | PSNR | SSIM | PSNR | SSIM |
| Chen *et al.* [Chen et al. 2022b] | 37.32 | 0.97 | 29.37 | 0.88 | 20.10 | 0.85 | 28.93 | 0.90 |
| WGWS [Zhu et al. 2023] | 38.94 | 0.98 | 33.64 | 0.93 | 29.46 | 0.85 | 34.01 | 0.92 |
| TransWeather [Valanarasu et al. 2022] | 33.64 | 0.93 | 29.16 | 0.82 | 17.33 | 0.82 | 26.71 | 0.86 |
| **TransWeather + Art (Ours)** | **38.58** | **0.98** | **30.06** | **0.91** | **20.07** | **0.88** | **29.57** | **0.92** |

**Table 10: Comparative results with 5 distinct tasks of all-in-one restoration.**

| Methods | SOTS | | Rain100L | | BSD68 | | GoPro | | LOL | | Average | |
|---|---|---|---|---|---|---|---|---|---|---|---|---|
| | PSNR | SSIM | PSNR | SSIM | PSNR | SSIM | PSNR | SSIM | PSNR | SSIM | PSNR | SSIM |
| TAPE [Liu et al. 2022] | 29.67 | 0.904 | 22.16 | 0.861 | 30.18 | 0.855 | 24.47 | 0.763 | 18.97 | 0.621 | 25.09 | 0.801 |
| AirNet [Li et al. 2022] | 32.98 | 0.951 | 21.04 | 0.884 | 30.91 | 0.882 | 24.35 | 0.781 | 18.18 | 0.735 | 25.49 | 0.846 |
| **AirNet + Art (Ours)** | **33.67** | **0.959** | **21.91** | **0.889** | **31.01** | **0.885** | **25.07** | **0.789** | **20.33** | **0.745** | **26.40** | **0.853** |
| Transweather [Valanarasu et al. 2022] | 29.43 | 0.905 | 21.32 | 0.885 | 29.00 | 0.841 | 25.12 | 0.757 | 21.21 | 0.792 | 25.22 | 0.836 |
| **Transweather + Art (Ours)** | **29.93** | **0.908** | **22.09** | **0.891** | **29.43** | **0.843** | **25.61** | **0.776** | **21.99** | **0.811** | **25.81** | **0.846** |

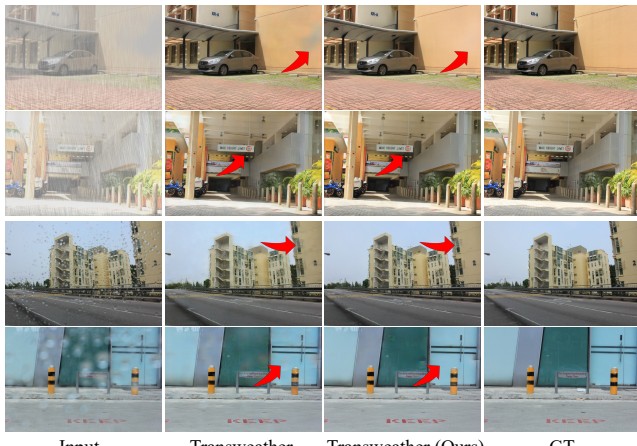

Input      Transweather      Transweather (Ours)      GT

**Figure 5: Visual results of Transweather and ours. More accurate and better quality images are obtained from the retrained Transweather.**

demonstrate the effectiveness of our Art method in enhancing the performance of both AirNet and Transweather models across all five tasks. It is worth noting that while the retrained AirNet model outperforms the retrained Transweather model on average, the performance gap between the two models is reduced compared to their original versions. This suggests that our Art method is effective in improving the performance of different architectures, making them more competitive in all-in-one image restoration tasks.

*Discussion*. Our experiments demonstrate the effectiveness of the proposed Art optimizing strategy for all-in-one image restoration across various degradation scenarios. Art consistently enhances the performance of state-of-the-art models, including AirNet, PromptIR, TransWeather, and MioIR, outperforming both vanilla mixed training and sequential training strategies. Notably, the retrained AirNet's ability to match or surpass the original PromptIR

(Table 7) highlights a critical insight: current limitations in all-in-one image restoration are not solely due to model capacity or architecture, but also stem from suboptimal training strategies. This finding emphasizes the importance of addressing the implicit effects of multiple tasks in the field.

## 4 LIMITATIONS

While our proposed Art approach marks a significant advancement in all-in-one image restoration, it is not without limitations. Our findings reveal uneven performance improvements across different tasks, as evidenced in Tables 6 and 7. Although Art substantially enhances the performance of models like Uformer and AirNet, the degree of improvement varies among tasks. This variability likely stems from the inherent complexities and diverse characteristics of the tasks involved, highlighting the need for further research.

## 5 CONCLUSION

This paper introduces a novel Multi-Task Balanced Learning (Art) approach for all-in-one image restoration, addressing the challenges of imbalanced learning status, and potential conflicts among multiple restoration tasks. Art explicitly formulates the restoration process as a multi-task learning problem, introducing task-specific rebalancing coefficients, a local convergence rate, and a novel global correction weight to equilibrate dynamics optimization and mitigate inter-task conflicts. Extensive experiments demonstrate the versatility and generalizability of Art across various task settings and model architectures, highlighting its potential to become a standard optimization strategy in the field of all-in-one image restoration. The insights and techniques introduced in this work lay the foundation for future research in multi-task learning and have the potential to revolutionize the approach to all-in-one image restoration and related problems in computer vision.

## 6 ACKNOWLEDGEMENT

The research was supported by the National Natural Science Foundation of China (U23B2009).

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
