# OpenReview forum: "Harmony in Diversity: Improving All-in-One Image Restoration via Multi-Task Collaboration"
_acmmm.org/ACMMM/2024/Conference — MM2024 Poster_

### Official Review · Reviewer_tBYE · 2024-05-20

**Rating:** 4
**Confidence:** 4

**Summary:**

The motivation of this paper is both interesting and meaningful, exploring the relationships and potential conflicts between different tasks, proposing a plug-and-play Art strategy, and presenting an efficient loss function. Extensive experiments have demonstrated the effectiveness of the proposed modules.

**Strengths:**

1. The motivation for exploring the relationship in multi-tasks is meaningful and novel.
2. The proposed effective task-specific loss can balance the individual task training process.
3. The experimental results are impressive.

**Limitations:**

1. Some exciting all-in-one image restoration methods test single-task performance, I am curious about the performance of this article (although this is not closely related to the motivation of the paper).
2. The parameter, FLOPS, and inference time comparison are missed.

**Suitability:**

3

---

### Official Review · Reviewer_78WR · 2024-05-24

**Rating:** 4
**Confidence:** 3

**Summary:**

This paper proposes a active-reweighting strategy for all-in-one image restoration. It redefine the all-in-one image restoration tasks as a multi-task learning problem, and reweight the coefficients for each tasks with local convergence rate and global correction weight. The experiment results on a diverse range of all-in-one image restoration settings demonstrate the effectiveness of the proposed method.

**Strengths:**

1. The motivation to mitigate the potential conflicts among various restoration tasks is interesting.

2. The proposed active reweighting method with local convergence rate and global correction weight is straightforward and reasonable.

3. The experiment results on a diverse range of all-in-one image restoration settings and methods demonstrate the effectiveness of the proposed method.

**Limitations:**

The main problem about the proposed method is that the exploration about the conflicts between different restoration tasks is not very clear. The authors claim that the proposed method is designed to harmonize the optimization of multiple restoration tasks. However, the calculation of the local convergence rate and the global correction weight seems have no direct relation with the conflicts between different restoration tasks. The author needs to explain how the active reweighting harmonizes the optimization of different restoration tasks.

**Suitability:**

2

---

### Official Review · Reviewer_6uBJ · 2024-05-24

**Rating:** 4
**Confidence:** 3

**Summary:**

The paper introduces a novel approach called Art (Active-Reweighting Strategy) designed to enhance multi-task learning for image restoration. The proposed method addresses the common issue of inconsistent optimization rhythms among various image restoration tasks, such as denoising, deblurring, deraining, and dehazing. Art harmonizes the optimization process by introducing a local convergence rate and a global correction weight to balance the training dynamics across different tasks. This strategy is validated through extensive experiments, showing significant improvements in performance when integrated into existing models like AirNet and TransWeather.

**Strengths:**

This paper proposes an innovative approach to addressing multi-task optimization conflicts in image restoration. The active-reweighting strategy (Art) effectively balances the optimization process, leading to consistent performance improvements across different tasks. The method's plug-and-play nature allows it to be easily incorporated into existing models, demonstrating its versatility and robustness. Extensive experimental validation supports the efficacy of Art, showing substantial performance gains in terms of PSNR across various image restoration scenarios.

**Limitations:**

More tests on real-world datasets need to be conducted to demonstrate the practical applicability of the proposed method.

**Suitability:**

2

---

### Official Review · Reviewer_mp6w · 2024-05-28

**Rating:** 3
**Confidence:** 3

**Summary:**

This paper introduces a new multi-task balance learning method named Art, which is used to improve the performance of integrated image recovery. It constructs the recovery process as a multi-task learning problem, balancing dynamic optimization and mitigating inter-task conflicts by introducing task-specific rebalancing coefficients, local convergence rates, and global correction weights. Experiments show that Art works well on image restoration tasks.

**Strengths:**

1.The introduction of Art showcases a novel method for enhancing multi-task image restoration, offering a fresh perspective in the field.
2.Art's design as an easily integrable optimization strategy allows it to be a versatile tool for various restoration tasks. And the method significantly improves upon existing models, indicating a substantial advancement in image restoration capabilities.
3.The paper provides extensive experiments that validate the effectiveness of Art across different datasets and models, adding to its credibility.

**Limitations:**

1.Although an effective optimization strategy is proposed in this paper, the theoretical analysis of complex relationships and different characteristics among tasks is not deep enough, and more theoretical analysis is needed to explain why it works.
2.There are some problems in the organization of the article. The Settings of the ablation experiment in Table 1 are not clear enough. Authors should make a good arrangement. Why is there an obvious low value (14.77) in the table, is the network not converging.
3.The datasets for all-in-one task may be unbalanced. The method might not perform optimally on all-in-one task with unbalanced datasets without further adjustments to handle class imbalances.

**Suitability:**

2

---

### Meta-Review · Area_Chair_Xvzn · 2024-07-02

**Recommendation:** Accept (Poster)
**Confidence:** 5

**Metareview:**

This paper proposes a new multi-task balance learning method to improve the performance of integrated image recovery, which provides a new perspective in the field. It constructs the recovery process as a multi-task learning problem, balancing dynamic optimization and mitigating inter-task conflicts by introducing task-specific rebalancing coefficients, local convergence rates, and global correction weights. Experimental results demonstrate that the proposed method performs well on various image restoration tasks. After rebuttal, this paper receives 4 positive reviews, so I recommend the acceptance of this paper.